# High-Performance Control Strategy for Low-Speed Torque of IPMSM in Electric Construction Machinery

**Zhongshen Li** [1,2], **Qihuai Chen** [1,2], **Yongjie Chen** [1,2], **Tianliang Lin** [1,2,*], **Haoling Ren** [1,2] **and Wen Gong** [3]

1   College of Mechanical Engineering and Automation, Huaqiao University, Xiamen 361021, China
2   Fujian Key Laboratory of Green Intelligent Drive and Transmission for Mobile Machinery, Xiamen 361021, China
3   Shanghai Institute of Special Equipment Inspection and Technical Research, Shanghai 200062, China
*   Correspondence: ltl@hqu.edu.cn

**Abstract:** Electric construction machinery with zero emission and high efficiency is considered to be a main development trend. An electric motor (EM) in electric construction machinery often needs to work at low speed or even in stalling conditions with large torque value and high work efficiency. In this paper, based on vector control of the maximum torque per ampere (MTPA) of an internal permanent magnet synchronous motor (IPMSM), a voltage and current composite observation algorithm was studied to improve IPMSM control performance at low speed. By establishing the observer model, the amplitude limited compensation for the voltage flux observation method was used to observe the EM stator flux. After being combined with the current observation method and the stator current obtained by coordinate transformation, the EM parameters in real-time can be identified for MTPA. Experimental research was carried out. The results show that the algorithm improved the speed control characteristic and output torque per unit current of the EM under low-speed working conditions.

**Keywords:** construction machinery; energy saving; electrification; vector control; low speed; parameter identification





## 1. Introduction

Electric construction machinery can achieve zero emission by replacing the engine with an electric motor (EM). Energy efficiency can be improved to a great extent, which is considered to be one of the main trends for construction machinery [1–4]. At present, some research has been conducted on the electrification of construction machinery. However, most of them just used the EM to replace the engine and simulate the engine working mode. The traveling construction machinery still realizes the matching and adaptation between the traveling load and the power train system by retaining a hydraulic torque converter. Due to the low efficiency of the hydraulic torque converter, the energy efficiency of the traveling power train system is unsatisfactory. Canceling the hydraulic torque converter and using the EM combined with a transmission to directly drive the traveling mechanism can effectively improve the system's efficiency. However, traveling construction machinery often needs the transmission system to work at a low speed (The speed is lower than 10% of the rated speed and 3% of the maximum speed) or even in stalling (the speed is 0 rpm) with large torque condition. Therefore, it is necessary to carry out deep research on the control method for the EM in low-speed driving conditions of a traveling transmission system [5].

Currently, some research has been conducted on the control of the low-speed performance of EM control. Chen et al. studied the maximum torque per ampere (MTPA) for an internal permanent magnet synchronous motor (IPMSM) that exhibits non-sinusoidal back-electromotive-force. An optimal current was presented as a function of current harmonics [6]. Han et al. improved the performance of an IPMSM based on space voltage

vector modulation schemes; a highly accurate online method to find the proper MTPA angles was presented [7]. Sun et al. introduced a novel virtual signal injection-based control method of MTPA for a IPMSM [8]. Djeriou et al. proposed an original control method based on the grey wolf (GW) algorithm, which could quickly optimize the control process and reduce the EM speed fluctuation at a low speed [9]. Elsonbaty et al. studied on torque ripple optimal control of a hybrid excitation synchronous motor at a low speed [10]. Zhu et al. proposed an EM low-speed dead zone compensation control algorithm based on the combination of a neural network band-pass filter and extended Kalman filter. The algorithm could effectively reduce the harmonic pulsation when the EM was running at a low speed, thus improving the utilization rate of DC bus voltage [11]. Chen et al. proposed a direct torque control (DTC) algorithm for a IPMSM of hybrid excavator. The algorithm was shown t o improve the low-speed torque ripple brought about by the traditional DTC, reduce the influence of DC bus voltage on the torque control, and improve the dynamic response ability of the EM [12]. Fu et al. proposed a sliding mode control algorithm based on vector control, which made the EM have better anti-interference ability when running at low speed [13]. Bobtsov et al. proposed a stator flux observer and load torque estimation method for non-salient permanent magnet synchronous motor (PMSM), which identified the stator flux value of the EM through the combination of linear time invariant (LTI) filter and linear regression, and then combined with the actual control to reduce the low-speed torque ripple of the EM and enhance the low-speed running stability [14]. Pulvirenti et al. proposed an identification algorithm for online identification of parameters such as stator resistance and permanent magnet (PM) flux linkage of three-phase open winding PMSM. The algorithm could make the EM output torque at a stable low speed. Meanwhile, it could also use the identified parameters to establish the EM temperature model and predict the thermal state [15]. Shi et al. proposed an EM parameter identification algorithm based on the extended Kalman filter, which could better control the speed accuracy of EM at a low speed [16]. Wang et al. proposed a model reference adaptive parameter identification system based on Popov hyper stability theory and used the algorithm to predict the current model. The algorithm could reduce the vibration torque at a low speed [17].

Currently, the research on the low-speed performance of EM mainly used the state parameters to measure the physical parameters, and then combined them with the DTC or the vector control to reduce the torque ripple and the overshoot in low-speed working conditions. However, the operation efficiency of EM under circumstance of low-speed or stalling was not considered. Aiming at the traveling system in construction machinery, a high-performance control for an IPMSM in the low-speed situation of the EM direct drive traveling system was studied in this paper. A voltage and current composite observation algorithm based on the MTPA was proposed for IPMSM control.

## 2. Voltage and Current Composite Observation Algorithm Framework

The schematic diagram is shown in Figure 1. The voltage and current composite observation algorithm can be divided into three parts. They are the EM temperature prediction model, the voltage observation model, and the current observation model, respectively. The EM temperature prediction model is a combination of the EM stator winding temperature model and the rotor PM temperature model. The voltage observation model is an EM mathematical model, which is controlled by the combination of the stator voltage and the current in the $\alpha$-$\beta$ coordinate system. The current observation model was based on the stator current in the d-q coordinate system.

First, the real-time stator resistance value and the PM flux linkage amplitude of the EM were obtained through the temperature model. Then, the stator resistance value and the PM flux linkage value were sent into the voltage observation and the current observation model. Through mathematical analysis, the flux component of the stator flux in the $\alpha$-$\beta$ coordinate system was obtained. Finally, through Park transformation, the flux component of the stator flux in the d-q coordinate system was obtained, and then sent to the current observation. Using the mathematical relationship between the stator current value and

the EM inductance in the d-q coordinate system, the real-time d-q inductance of EM was calculated for MTPA.

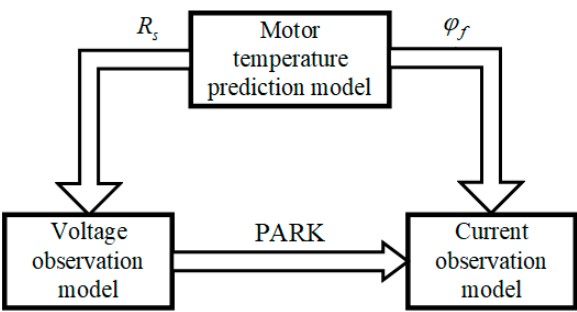

**Figure 1.** Schematic diagram of voltage and current composite observation algorithm.

## 3. Torque Improvement Model Design of IPMSM

### 3.1. MTPA Control

MTPA control is intended to output the corresponding maximum torque with the most appropriate d-q axis current ratio by controlling the output of each unit current of the EM.

The torque equation and the current equation of EM can be deduced as

$$T_e = \frac{3}{2} n_p \big[ \varphi_f + (L_d - L_q) i_d \big] i_q \tag{1}$$

$$i_s = \sqrt{i_d^2 + i_q^2} \tag{2}$$

where $T_e$ is the output torque of EM. $n_p$ is the number of pole pair of EM. $\varphi_f$ is the PM flux linkage of different temperature. $L_d$ and $L_q$ are the d-q axis inductance of EM, respectively. $i_d$ and $i_q$ are the d-q axis current of EM, respectively. $i_s$ is the amplitude of the current.

In order to make the unit current of the motor correspond to the maximum output torque, Equations (1) and (2) can be combined together with Lagrange's theorem:

$$H = \sqrt{i_d^2 + i_q^2} + \lambda \left\{ T_e - \frac{3}{2} n_p \big[ \psi_f + (L_d - L_q) i_d \big] i_q \right\} \tag{3}$$

where $H$ is the Lagrange operator and $\lambda$ is the weight coefficient.

The partial derivative of parameter $H$ can be obtained as

$$\begin{cases} \frac{\partial H}{\partial i_d} = \frac{i_d}{\sqrt{i_d^2 + i_q^2}} + \frac{3}{2} \lambda n_p (L_q - L_d) i_q = 0 \\ \frac{\partial H}{\partial i_q} = \frac{i_d}{\sqrt{i_d^2 + i_q^2}} + \frac{3}{2} \lambda n_p (L_q - L_d) i_d = 0 \\ \frac{\partial H}{\partial i_q} = T_e - \frac{3}{2} n_p \big[ \psi_f + (L_d - L_q) i_d \big] i_q = 0 \end{cases} \tag{4}$$

The most reasonable d-q axis current can be obtained as

$$\begin{cases} i_d = \frac{\psi_f}{2(L_d - L_q)} - \sqrt{\frac{\psi_f^2}{4(L_d - L_q)^2} + i_q^2} \\ i_q = \sqrt{i_s^2 - i_d^2} \end{cases} \tag{5}$$

Set the angle between $i_s$ and $i_d$ as $\beta$. According to the coordinate transformation relationship, the following formula can be obtained as

$$\begin{cases} i_d = \cos \beta * i_s \\ i_q = \sin \beta * i_s \end{cases} \tag{6}$$

Through Equations (5) and (6), $\beta$ can be obtained as

$$\beta = \text{acos} \frac{-\varphi_f + \sqrt{\varphi_f^2 + 8(L_d - L_q)^2 i_s^2}}{4(L_d - L_q)i_s} \tag{7}$$

The control block diagram is shown in Figure 2.

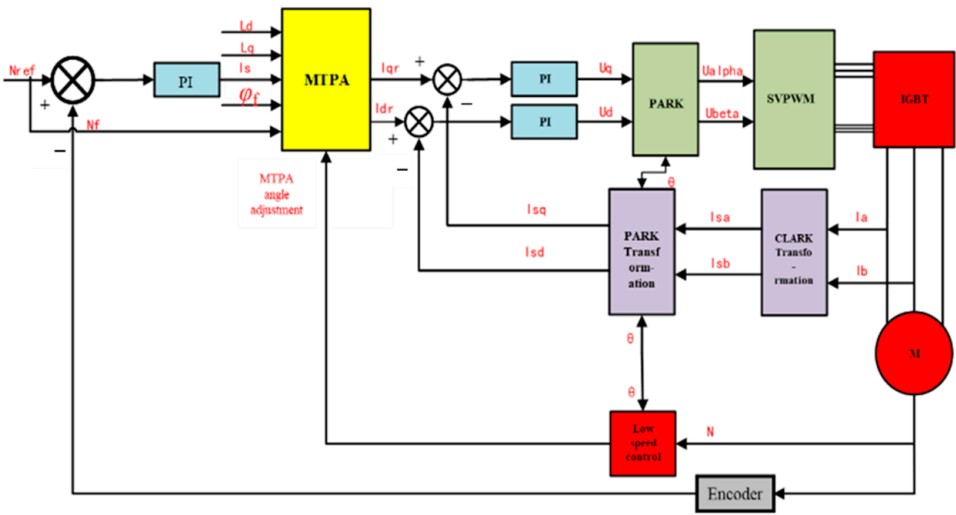

**Figure 2.** Block diagram of MPTA.

### 3.2. EM Temperature Prediction Model

The EM temperature prediction model aims to obtain the variation of EM flux linkage and the stator resistance through temperature changes. The model between the resistance of the EM stator and the amplitude of PM flux linkage varied with temperature can be expressed as

$$\begin{cases} R_s = R_0[1 + \alpha(t - 25)] \\ \varphi_f = \varphi_0[1 + \alpha_f(t - 25)] \end{cases} \tag{8}$$

where $R_s$ is the resistance of stator of different temperature. $R_0$ is the resistance of stator at 25 °C. $\varphi_0$ is the PM flux linkage at 25 °C. $\alpha$ is the resistance temperature coefficient of copper. $\alpha_f$ is the reversible temperature coefficient of PM materials.

### 3.3. Voltage Observation Model

The voltage observation model is mainly used to observe the component of the EM stator flux in the $\alpha$-$\beta$ coordinate system. The voltage observation model is based on the flux observer in the DTC of PMSM. The function of flux observer in DTC is to analyze and calculate the sampled values of the stator voltage and current. The stator flux vector of EM can be given as

$$\begin{cases} \varphi_\alpha = \int (u_\alpha - R_s i_\alpha) dt \\ \varphi_\beta = \int (u_\beta - R_s i_\beta) dt \end{cases} \tag{9}$$

where $\varphi_\alpha$ and $\varphi_\beta$ are components of stator flux linkage in the $\alpha$-$\beta$ coordinate system, respectively. $u_\alpha$ and $u_\beta$ are components of stator voltage in the $\alpha$-$\beta$ coordinate system, respectively. $i_\alpha$ and $i_\beta$ are components of stator current in the $\alpha$-$\beta$ coordinate system, respectively

The traditional flux observer was integrated into the back electromotive force (EMF) to obtain the stator flux value. The flux observer is only related to the real-time acquisition accuracy of EM current and stator resistance $R_s$. However, when the EM was in low-speed conditions, the amplitude of back EMF was shown to be small and the amplitude of stator current increased when the load increased. At this time, the drop of voltage caused by the

EM stator resistance was impossible to ignore, leading to the increase of the observation error of the flux observer. Meanwhile, the traditional flux observer directly integrated the back EMF to obtain the stator flux vector value. The direct integration easily produced a DC bias, which affected the accuracy of the stator flux observation. Therefore, to solve the problem of the traditional voltage flux observer in low-speed conditions, an amplitude limiting compensation was employed. The principle of voltage observation model based on amplitude limiting compensation mode is shown in Figure 3.

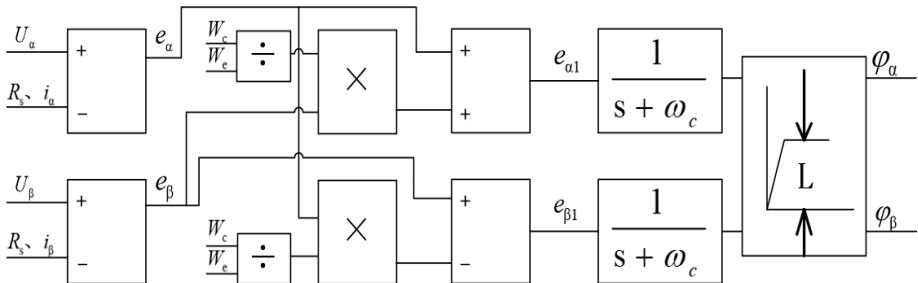

**Figure 3.** Amplitude limited compensation voltage observation model.

As can be seen, the voltage observation model based on the amplitude limiting compensation method is a multi-parameter coupled system, which uses the comprehensive effects of the voltage and current input, such as difference multiplication and speed feedback, and finally outputs stable and accurate $\psi_\alpha$ and $\psi_\beta$ through low-pass filter and amplitude limited compensation.

The back EMF of EM in $\alpha$-$\beta$ coordinate system can be expressed as

$$\begin{cases} e_\alpha = u_\alpha - R_s i_\alpha \\ e_\beta = u_\beta - R_s i_\beta \end{cases} \tag{10}$$

where $e_\alpha$ and $e_\beta$ are the back EMF components of EM in the $\alpha$-$\beta$ coordinate system, respectively.

The improved back EMF in the $\alpha$-$\beta$ coordinate system $e_{\alpha 1}$ and $e_{\beta 2}$ can be expressed as

$$\begin{cases} e_{\alpha 1} = e_\alpha + \frac{W_c}{W_e} \times e_\beta \\ e_{\beta 1} = e_\beta - \frac{W_c}{W_e} \times e_\alpha \end{cases} \tag{11}$$

where $e_{\alpha 1}$ and $e_{\beta 2}$ are improved back EMF components of EM in the $\alpha$-$\beta$ coordinate system, respectively. $W_c$ is the cutoff frequency. $W_e$ is the electric speed of EM. $L$ in Figure 2 is the limited amplitude of amplitude compensation module, which is used to compensate the possible amplitude reduction after low-pass filtering.

As can be seen in Figure 3 and Equation (10), when the EM runs at a low speed, the cutoff frequency is divided by the EM feedback electric speed in the front part of the amplitude limited compensation voltage observation model. It is then combined with the back EMF component $e_\alpha$ and $e_\beta$ to reduce the influence of DC component on the flux observation. The first-order inertial filter and the amplitude limiting compensation module in the back-end part are mainly used to compensate the output flux and keep its phase from distortion. At this time, the voltage observation model can be expressed as

$$\begin{cases} \psi_\alpha = \frac{1}{s+\omega_c} \times (e_\alpha + \frac{W_c}{W_e} \times e_\beta) \\ \psi_\beta = \frac{1}{s+\omega_c} \times (e_\beta - \frac{W_c}{W_e} \times e_\alpha) \end{cases} \tag{12}$$

where $\omega_c$ is the constant of first-order inertial filter.

### 3.4. Current Observation Model

The main function of the current observation model is to obtain the final d-q inductance value through Park transformation for the temperature prediction model together with the voltage observation model. The current observation model designed in this paper is consistent with the classical current flux linkage observer in IPMSM DTC. $\varphi_d$ and $\varphi_q$ can be obtained via Park transformation for the voltage observation model. Further, $\varphi_d$ and $\varphi_q$ can be fed into the mathematical model of the current observation model to obtain the *d-q* inductance. The coordinate system transformation from $\varphi_\alpha$ and $\varphi_\beta$ to $\varphi_d$ and $\varphi_q$ is shown in Figure 3.

The relationship between flux linkage from the α-β coordinate system to the d-q coordinate system can be deduced as

$$\begin{bmatrix} \varphi_d \\ \varphi_q \end{bmatrix} = \begin{bmatrix} \cos\theta & \sin\theta \\ -\sin\theta & \cos\theta \end{bmatrix} \begin{bmatrix} \varphi_\alpha \\ \varphi_\beta \end{bmatrix} \tag{13}$$

The mathematical model of current observation model can be expressed as

$$\begin{cases} \psi_d = L_d i_d + \psi_f \\ \psi_q = L_q i_q \end{cases} \tag{14}$$

Figure 4 and Equation (13) show that the voltage observation model outputs accurate $\varphi_\alpha$ and $\varphi_\beta$ for the current observation model. The stator current of the EM is collected in real time. Through Equation (13), the d-q inductance of the EM can be acquired.

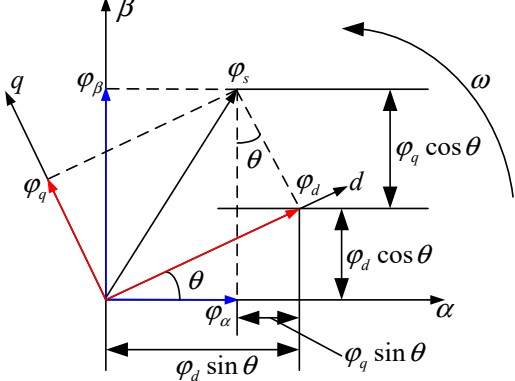

**Figure 4.** Schematic diagram of flux linkage coordinate system transformation.

The MPTA for an IPMSM based on the composite observation algorithm is given in Figure 5.

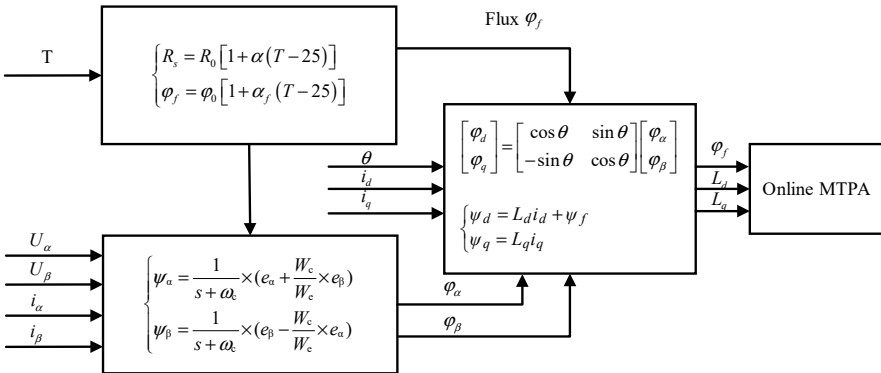

**Figure 5.** IPMSM MTPA based on the voltage and current composite observation algorithm.

## 4. Experimental Research

### 4.1. Experimental Platform

Experimental research was conducted. The experimental platform is given in Figure 6. Parameters of the tested IPMSM are given in Table 1.

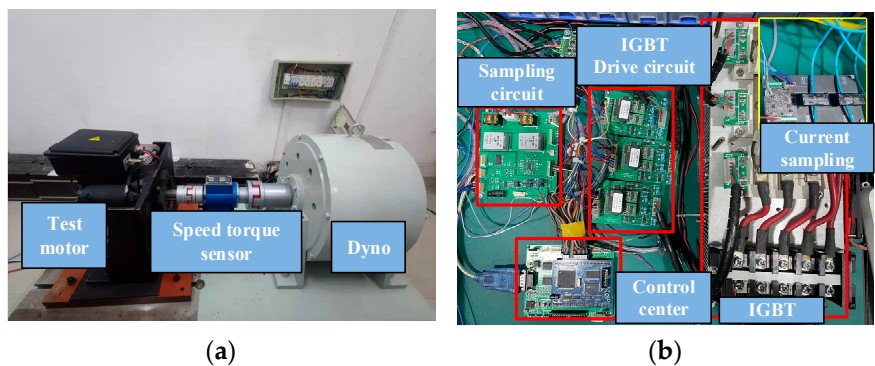

**(a)**        **(b)**

**Figure 6.** Experimental research hardware. (**a**) Experimental platform. (**b**) Invert of EM.

**Table 1.** Parameters of IPMSM.

| Rated Power (kW) | Rated Speed (rpm) | Rated Torque (N·m) | Rated Current (A) | $L_d$ (mH) | $L_q$ (mH) | Pole Pair |
|---|---|---|---|---|---|---|
| 9 | 1800 | 44 | 35 | 2.5 | 5.5 | 4 |

### 4.2. Test and Analysis of No-Load On-Line Parameter Identification

The EM ran at 120 rpm. The flux linkage component in the $\alpha$-$\beta$ coordinate system through composite observation algorithm under no load condition is given in Figure 6. As can be seen, the composite observation algorithm overcame the DC component interference and the amplitude limitation in the identification process. The EM flux linkage components in the $\alpha$-$\beta$ coordinate system were accurately identified by using the amplitude limiting compensation part. The EM flux linkage components in the $\alpha$-$\beta$ coordinate system converged to the steady state within 0.75 s. It can be seen from Figure 7b that the angle between the $\alpha$ axis flux linkage component and the $\beta$ axial flux linkage component was strictly different by 90°. The stable amplitude of the flux linkage component converged near 0.219 Wb. The flux linkage component obtained in Figure 7 is consistent with theoretical derivation, which verified the accuracy of the mathematical model built above.

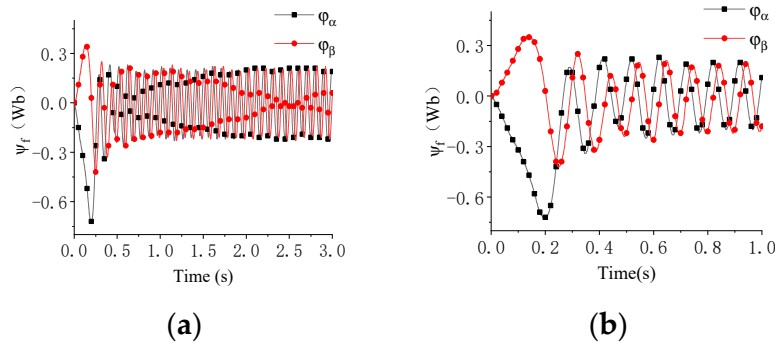

**(a)**        **(b)**

**Figure 7.** Test value of flux linkage component in the $\alpha$-$\beta$ coordinate system. (**a**) Flux linkage component in the $\alpha$-$\beta$ coordinate system. (**b**) Local amplification of flux linkage component in the $\alpha$-$\beta$ coordinate system.

The flux linkage component in the d-q coordinate system is given in Figure 8. As can be seen, the flux linkage component in the d-q coordinate system converged to a

certain value in short time, as well. The d-axis flux component converged to 0.219 Wb and the q-axis flux component converged to 0.01 Wb. Combined with the current model in the composite observation algorithm, when the EM ran without load, the stator current should be 0 A. The stator flux component of *d*-axis should be consistent with the PM flux $\varphi_f$. The q-axis stator flux component should be 0. However, due to certain inertia when the dynamometer did not run, the q-axis flux component was not 0. Therefore, the flux component identified by the test in Figure 7 is consistent with the theoretical derivation. The accuracy of the mathematical model of the algorithm was thus verified.

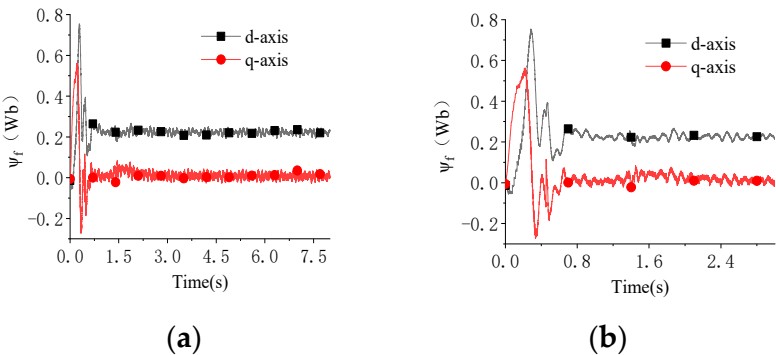

**Figure 8.** Test value of flux linkage component in the d-q coordinate system. (**a**) Flux linkage component in the d-q coordinate system. (**b**) Local amplification of flux linkage component in the d-q coordinate system.

The stator flux linkage calculated through flux linkage components of the α-β axis and the d-q axis is given in Figure 9. When the EM runs without load, the amplitude of stator flux was consistent with that of PM flux. The stator flux accurately converged to 0.219 Wb when the EM operated stably. Therefore, the PM flux test identification value of the test EM was found to be 0.219 Wb.

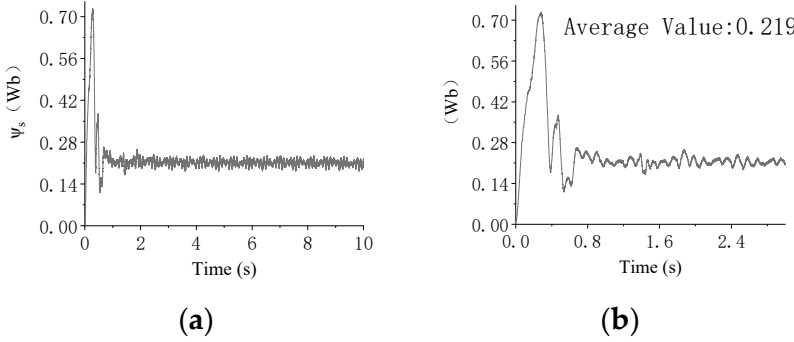

**Figure 9.** Identification value of stator flux linkage test. (**a**) Stator flux linkage. (**b**) Partial enlarged view of stator flux linkage.

Identification value of d-q axis inductance of the test EM during no-load is given in Figure 9. In the no-load test condition, due to the inertia of the dynamometer itself, the EM needs to provide a small output torque as the power source to drive the normal operation of the whole test bench. Therefore, as can be seen in Figure 10a, the q-axis current of the EM was not 0 after it became stable. It can be seen from Figure 10b,c that during the no-load operation of the test bench, the oscillation effect of the d-axis inductance identification value was obvious, and the q-axis inductance identification value converged to 0.006 H. Combined with the mathematical model of the composite observation algorithm and Figure 10a, it can be seen that in the no-load condition the $i_d$ oscillates near 0 A, so the calculated d-axis inductance identification value greatly fluctuated, and the q-axis inductance value was relatively stable.

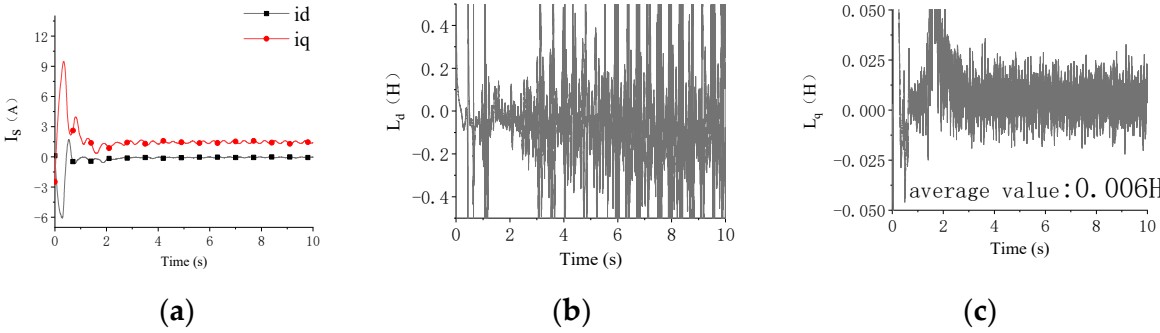

**Figure 10.** Identification values of stator current and d-q axis inductance of no-load EM. (**a**) Stator current of no-load EM. (**b**) Identification value of no-load d-axis inductance. (**c**) Identification value of no-load q-axis inductance.

Therefore, the no-load test verified the correctness of the mathematical model of the low-speed composite observation algorithm. Meanwhile, the PM flux linkage of the EM was obtained as 0.219 Wb.

### 4.3. Test and Analysis of On-Line Parameter Identification in Loaded Condition

The verification test in the loaded condition of the control algorithm was also carried out. The test conditions are shown in Table 2. Due to hardware limitations, no temperature sensor was installed inside the tested EM. Therefore, we decided to adopt the small load in the loaded test to reduce the influence of temperature on EM parameters. The influence of temperature on various parameters such as PM flux of the EM was ignored.

**Table 2.** Loaded test conditions.

| Target Torque (N·m) | Target Speed (r/min) | Time (s) |
| --- | --- | --- |
| 7 | 120 | 15~30 |
| 14 | 120 | 30~50 |

The stator current and the flux linkage component in the d-q coordinate system through voltage observation are given in Figure 11. During 15~30 s, $i_d$ of the EM is −0.4 A, $i_q$ was 5 A, the average value of d-axis flux linkage was 0.218 Wb, and the average value of q-axis flux linkage was 0.031 Wb. During 30–50 s, $i_d$ of the EM was −1.3 A, $i_q$ was 11 A, the average value of d-axis flux linkage was 0.217 Wb, and the average value of q-axis flux linkage was 0.065 Wb.

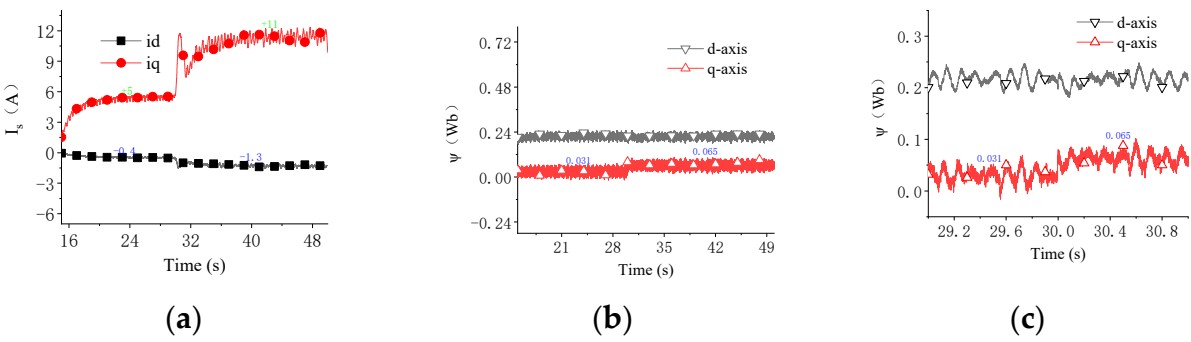

**Figure 11.** Identification value of stator current and d-q axis flux linkage component of loaded EM. (**a**) Stator current of loaded EM. (**b**) Loaded d-q axis flux linkage. (**c**) Local amplifier of loader d-q axis flux linkage.

The key value of the flux linkage identification algorithm was obtained in the experiment of the loaded EM flux linkage identification. The default value of the flux linkage

identification algorithm was also obtained in the experiment of the loaded EM flux linkage identification. As shown in Figure 12, during 15~30 s, the stator flux $\varphi_s$ was 0. 25 Wb. The average inductance of d-axis was 0.0025 H. The average inductance of q-axis was 0.006 H. During 30~50 s, the average value of the stator flux linkage $\varphi_s$ was 0.2268 Wb. The average inductance of d-axis was 0.0025 H. The average inductance of q-axis was 0.006 H. To sum up, the composite observation algorithm can identify the parameters of the EM under the actual load condition without knowing the accurate initial parameter values of the EM, and the parameter values can converge to a stable value, which can thus verify the feasibility and accuracy of the control algorithm.

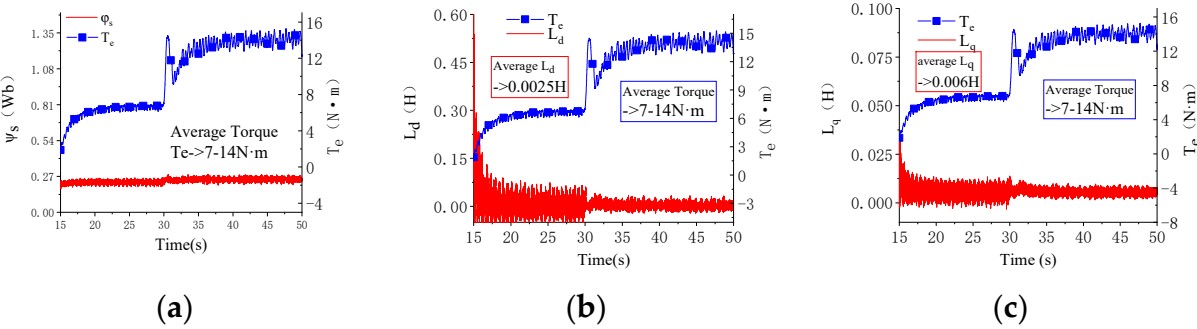

(**a**)                                  (**b**)                                  (**c**)

**Figure 12.** Identification value of loaded EM parameter test. (**a**) Stator flux linkage of loaded EM $\varphi_s$. (**b**) Identification value of inductance of d-axis for loaded EM. (**c**) Identification value of inductance of q-axis for loaded EM.

Through the no-load and loaded tests of the composite observation algorithm, the parameters during the operation of the EM were identified. Compared with the manufacturer's parameters, the results shown in Table 3 can be obtained. It was found that under the condition, the initial value of EM parameters was unknown. Moreover, there were some differences between the parameters identified by the composite observation algorithm and the manufacturer's parameters. Therefore, the comparison test between the manufacturer's parameters and the actual variable load of the identified parameters will be carried out to verify the reasonability of the identified parameters, and whether the voltage current composite identification algorithm proposed in this paper can make the EM output greater torque under the same working conditions.

**Table 3.** Comparison of parameters.

|  | **Manufacturer Parameters** | **Identification Value** |
|---|---|---|
| Flux linkage (Wb) | 0.203 | 0.219 |
| $L_d$ (mH) | 2.5 | 2.5 |
| $L_q$ (mH) | 5.5 | 6 |

*4.4. Comparative Test Analysis of Parameter Identification MTPA Control and Fixed Parameter MTPA Control*

The MTPA control was constructed by using the parameters of the EM manufacturer. When the initial parameter value of the test EM was unknown, the voltage and current composite observation algorithm was used to carry out the no-load and loaded tests. The parameters of the EM were identified, but there were differences with the manufacturer parameters. Next, the identified EM parameters are combined into the MTPA control to form a novel MTPA control, which was compared with the original fixed parameter MTPA control. The test conditions are shown in Table 4.

**Table 4.** Comparison test conditions.

|  | Fixed Parameter | Novel Control |
|---|---|---|
| Target speed (r/min) | 120 | 120 |
| Dynamometer given torque (N·m) | 0~7~14 | 0~7~14 |
| Time (s) | 0~15~30~45 | 0~15~30~45 |

The speed comparison diagram obtained from the test of the MTPA based on composite observation algorithm and the fixed parameter MTPA control are given in Figure 13. It can be seen that the parameter identification MTPA control had a better starting performance than the fixed parameter MTPA control. The starting time of MTPA based composite observation algorithm was shortened by 0.2 s compared with that of the fixed parameter MTPA control. The starting process was more stable and the speed fluctuation was less. Meanwhile, when the EM was subject to external disturbance during low-speed operation, the parameter identification MTPA control had a stronger anti-interference ability and kept the EM running according to the original setting state. Therefore, the EM parameters identified by the composite observation algorithm in the previous paper were found to be reasonable and effective.

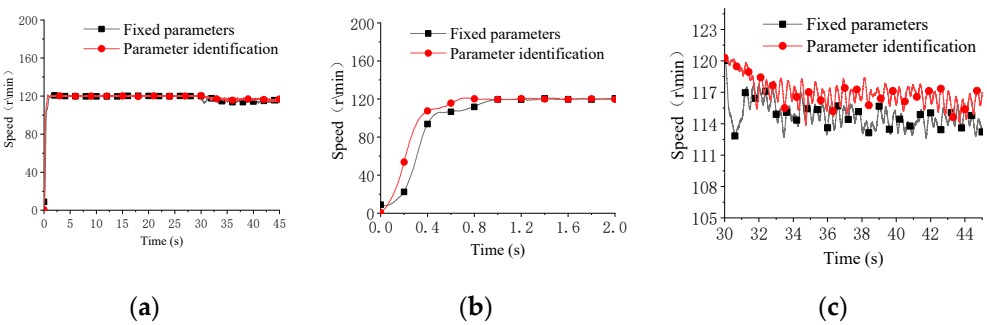

(**a**)        (**b**)        (**c**)

**Figure 13.** Comparison of EM speed between fixed parameter MTPA control and parameter identification MTPA control. (**a**) Speed comparison. (**b**) Partial enlarged view of speed comparison front section. (**c**) Partial enlarged view of rear section of speed comparison.

The comparison diagram of the EM stator current obtained from the comparison test between the MTPA based on the composite observation algorithm and the fixed parameter MTPA are given in Figure 14. It can be seen that compared with the fixed parameter MTPA control, the parameter identification MTPA control changed the MTPA control angle during the operation of the EM with the identified EM parameters, so that the current components in the d-q coordinate system of the parameter identification MTPA control was different from that of the fixed parameter MTPA control.

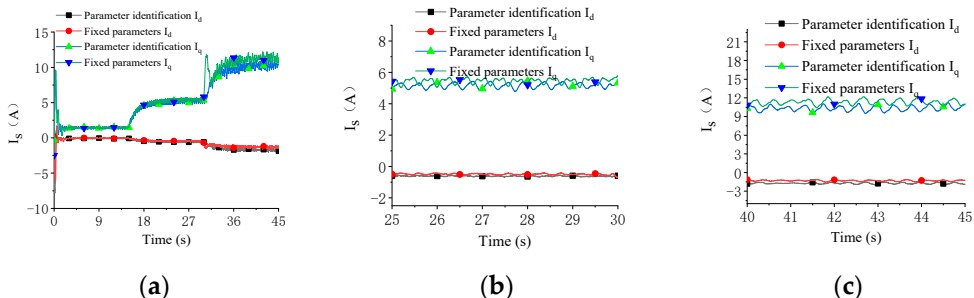

(**a**)        (**b**)        (**c**)

**Figure 14.** Comparison of current components in the d-q coordinate system between fixed parameter MTPA control and parameter identification MTPA control. (**a**) Comparison of stator current. (**b**) Partial enlarged view of middle section. (**c**) Partial enlarged view of end section.

The output torque and the stator current obtained from the comparison test of the fixed parameter MTPA control and the MTPA based composite observation algorithm are given in Figures 15 and 16, respectively. It can be seen form Figure 15 that when the dynamometer gave the same load torque, the output torques of the EM under the two control modes were consistent and did not change with the variation of load torque. However, the parameter identification MTPA control had less output torque ripple and better anti-torque ripple performance at the moment of load sudden change. It can also be seen from Figure 16 that during 0~15 s, the dynamometer did not work, and the stator current amplitudes of the tested EM were consistent in the two-control method. During 15~30 s, 7 N·m was loaded for EM by the dynamometer. At this time, the EM output torque was consistent under the two-control method, but the stator current under parameter identification MTPA control was reduced by 0.4 A compared with the fixed parameter MTPA control. During 30~45 s, 14 N·m was loaded for the EM. At this time, the output torques of the two-control method were the same, but the stator current under the parameter identification MTPA control was reduced by 1.1 A compared with the fixed parameter MTPA control.

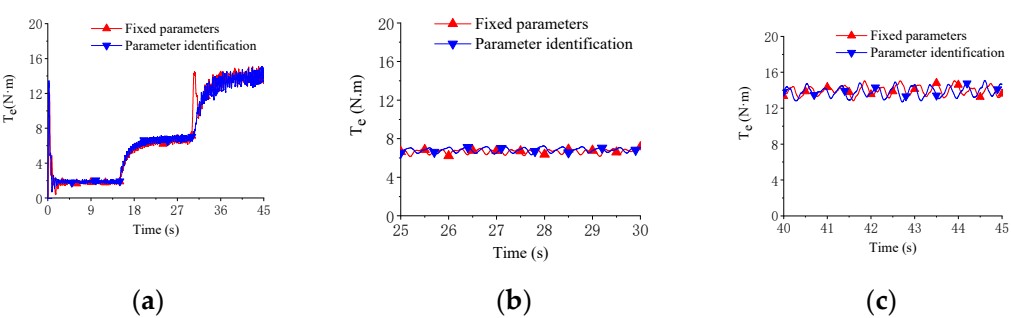

$$\text{(a)} \qquad\qquad\qquad \text{(b)} \qquad\qquad\qquad \text{(c)}$$

**Figure 15.** Comparison of EM output torque between fixed parameter MTPA control and parameter identification MTPA control. (**a**) Comparison of torque. (**b**) Partial enlarged view of middle section. (**c**) Partial enlarged view of end section.

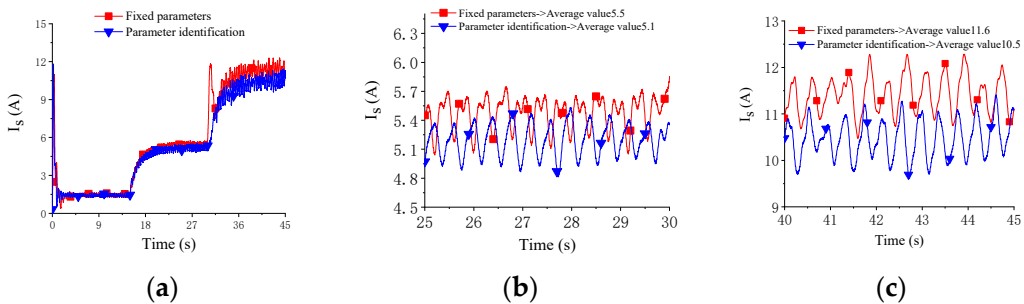

$$\text{(a)} \qquad\qquad\qquad \text{(b)} \qquad\qquad\qquad \text{(c)}$$

**Figure 16.** Comparison of stator current between fixed parameter MTPA control and parameter identification MTPA control. (**a**) Comparison of stator current. (**b**) Partial enlarged view of middle section. (**c**) Partial enlarged view of end section.

Therefore, the parameters of the EM obtained by using the voltage and the current composite observation algorithm in the EM no-load and loaded tests were rational, which verified the feasibility and correctness of the voltage and current composite identification algorithm proposed in this paper. In addition, it was further proven that the voltage current composite observation algorithm could optimize the operation performance of the EM when the EM was in a low-speed condition. At the same time, the utilization rate of stator current could be increased by 9.5%.

## 5. Conclusions

(1) Electric construction machinery is considered to be an important trend in the future. However, electric construction machinery has more stringent requirements for EM

control. It is often necessary for the EM to work under the condition of low speed, large torque, and high efficiency.

(2) Based on the vector control of the MTPA for an IPMSM, we studied a voltage and current composite observation algorithm. By establishing the EM temperature model and observing the temperature change during the low-speed condition of the EM, the real-time stator resistance value and the PM flux linkage of the EM were observed, and then sent into the voltage observation model. During the operation of the EM, the amplitude limited compensation voltage flux linkage observation method was used to observe the stator flux, and then combined with the current observation model. The parameters in the real-time operation of the EM were identified.

**Author Contributions:** Conceptualization, Z.L. and T.L.; methodology and software, Y.C.; formal analysis, Q.C. and W.G.; investigation, H.R. and Y.C.; writing—original draft preparation, Q.C. and H.R. All authors have read and agreed to the published version of the manuscript.

**Funding:** This research was funded by National Natural Science Foundation of China (Grant No. 51875218 and 52175051), Key Projects of Natural Science Foundation of Fujian Province (Grant No. 2021J02013), Collaborative Innovation Platform of Fuzhou-Xiamen-Quanzhou Independent Innovation Demonstration Area (Grant No. 3502ZCQXT202002), Fujian University Industry University Research Joint Innovation Project Plan (Grant No.2022H6007) and Shanghai Municipal Administration for Market Regulation (Grant No.2021-26).

**Institutional Review Board Statement:** Not applicable.

**Informed Consent Statement:** Not applicable.

**Data Availability Statement:** Not applicable.

**Conflicts of Interest:** The authors declare no conflict of interest.

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
