# Peer review of "High-Performance Control Strategy for Low-Speed Torque of IPMSM in Electric Construction Machinery"

_machines, doi:10.3390/machines10090810_

Round 1

Reviewer 1 Report

First, I want to say that it is an interesting paper that required a lot of work, so congratulations!

Some observations:

Nonsense sentences (unfinished):

1.       “While the motor in electric construction machinery often needs to work at low speed or even stalling conditions with large torque value and high work efficiency which rarely encountered in industrial environments”!

2.       “The current observation model is a model in which the stator current of the motor in the d-q coordinate system”!

3.       “It can be seen from Figure 3 and equation (6) that when the voltage observation model outputs accurate φα and φβ for the current observation model”!

4.       “While, no-load test, loaded test and comparative test are carried out, respectively”!

I consider it would be better if the explanation of the algorithm (e.g., fig. 4) will start with the measured values!

Why rated power differs from rated torque x rated speed [rad/s]?

If it is possible, please specify the voltage!

It should be better if you will specify, what was the speed in the no-load test?

The statement, “… the angle between the α axis flux linkage component and the β axial flux linkage component is strictly different by 90 ° ” - can you show more accurate this angle? (How did you notice?) Which is its value?

In the conclusions caption, you refer to the temperature model of the motor and parameter recalculation, but in the test, you do not use this parameter. It is commendable to conclude on the tested models!

Author Response

The manuscript has been revised according to reviewers' comments.

Reviewer 2 Report

   In this paper, the authors propose a voltage and current composite observation algorithm to enhance the control performance of IPMSM motor for the applications of the construction machinery. To begin with, the overall structure of the paper is good, and the related background information of the research is adequately cited and presented. Then, the introduction of the design methodology is also quite well-organized accompanied by proper illustrations from the diagrams. Lastly, the experimental research is also presented to verify the proposed control algorithm.

        Meanwhile, I have a few questions/suggestions for this paper:

1    1. In Section 3, the authors have not presented sufficient information regarding the MTPA adopted in the torque control. Can this information be added and introduced in the revised version of the paper? In addition, I have a few suggested references to be included in this paper:

(1): Chen, Taowen, et al. "Maximum torque per ampere control of interior permanent magnet synchronous motor via optimal current excitation." 2019 IEEE Energy Conversion Congress and Exposition (ECCE). IEEE, 2019.

(2): Han, Zexiu, et al. "Improved online maximum-torque-per-ampere algorithm for speed controlled interior permanent magnet synchronous machine." IEEE Transactions on Industrial Electronics 67.5 (2019): 3398-3408.

(3): Sun, Tianfu, Jiabin Wang, and Xiao Chen. "Maximum torque per ampere (MTPA) control for interior permanent magnet synchronous machine drives based on virtual signal injection." IEEE Transactions on Power Electronics 30.9 (2014): 5036-5045.

2      2. The are some issues with the loaded test: First, the motor temperature prediction is ignored in the test, but it is an important module for the voltage and current composite observation algorithm. How can you verify the benefits of inclusion of this module in the actual application? Second, the load torque is too small compared to the actual application scenarios with high load torque, low speed requirements, how can you justify your control algorithm under such operation demands?

3       3. Do the main differences of the experimental results between the MTPA based on composite observation algorithm and the fixed parameter MTPA come from the parametric discrepancies of the motor parameters (Table 3)?

Author Response

(The authors gave the same response as above.)

Reviewer 3 Report

In the article under review, the authors proposed a strategy for improving the performance of an electric drive with an internal permanent magnet synchronous motor (IPMSM) and with vector control of maximum torque per ampere (MTPA), when operating at low speeds or even stalling conditions with large torque value.

The results of the presented studies can be useful to specialists in the field of electric drive control systems.

In the Introduction and the literature review, the prerequisites for conducting research are considered in sufficient volume, and the purpose of the work is formulated. In the main parts of the paper, the IPMSM output torque improvement concept proposed by the authors based on the combined voltage and current composite observation algorithm is described in sufficient detail. The results of experimental studies on a laboratory stand are presented, which generally confirmed the theoretical developments and the performance of the proposed algorithm.

During the review, I was left with a few questions and I drew attention to the following shortcomings:

  1. Please explain what does "low motor speed" mean? How many times is this lower than the nominal value?
  2. What is the range of speed control for the considered class of mechanisms? The answer to this question may clarify the relevance of the research, which is still unclear.
  3. Please explain what does “high performance control” mean? It seems to me that in the Introduction section it would be worth explaining this term in more detail.

In addition, the following edits must be made to the paper:

  1. In Figure 2 and others, and especially in Figure 3, there are too small inscriptions. For easy reading of the paper, they should be enlarged.

Author Response

(The authors gave the same response as above.)

Round 2

Reviewer 2 Report

The authors have largely addressed my questions/suggestions. Please double check any minor grammatical errors in the paper.

Author Response

The manuscript has been revised according to the reviewers' comments.
